Linguacodus: a synergistic framework for transformative code generation in machine learning pipelines

Trofimova Ekaterina 1 etrofimova@hse.ru
Sataev Emil 1
Ustyuzhanin Andrey 2 3
1 Faculty of Computer Science, Higher School of Economics , Moscow , Russia
2 IFIM, National University of Singapore , Singapore , Singapore
3 School of Computer Science & Engineering, Constructor University , Bremen , Germany
Kong Xiangjie
Electronic publication date: 2024 Sep 23
Publication date: 2024
Volume: 10
Electronic Location ID: e2328
Received 2024 May 1; Accepted 2024 Aug 22
Copyright: © 2024 Trofimova et al.
Copyright year: 2024
Copyright holder: Trofimova et al.
License: This is an open access article distributed under the terms of the Creative Commons Attribution License, which permits unrestricted use, distribution, reproduction and adaptation in any medium and for any purpose provided that it is properly attributed. For attribution, the original author(s), title, publication source (PeerJ Computer Science) and either DOI or URL of the article must be cited.
License URL: https://creativecommons.org/licenses/by/4.0/

Keywords: Automated code generation, Large language models, Machine learning pipelines

Funding: National Research University Higher School of Economics Russian Science Foundation: n°19-71-30020 The reported study utilized the supercomputer resources of the National Research University Higher School of Economics. The research leading to these results has received funding from the Russian Science Foundation under grant agreement n°19-71-30020. The funders had no role in study design, data collection and analysis, decision to publish, or preparation of the manuscript.

==============================
In the ever-evolving landscape of machine learning, seamless translation of natural language descriptions into executable code remains a formidable challenge. This article introduces Linguacodus, an innovative framework designed to tackle this challenge by deploying a dynamic pipeline that iteratively transforms natural language task descriptions into code through high-level data-shaping instructions. The core of Linguacodus is a fine-tuned large language model, empowered to evaluate diverse solutions for various problems and select the most fitting one for a given task. This article details the fine-tuning process and sheds light on how natural language descriptions can be translated into functional code. Linguacodus represents a substantial leap towards automated code generation, effectively bridging the gap between task descriptions and executable code. It holds great promise for advancing machine learning applications across diverse domains. Additionally, we propose an algorithm capable of transforming a natural description of an ML task into code with minimal human interaction. In extensive experiments on a vast machine learning code dataset originating from Kaggle, we showcase the effectiveness of Linguacodus. The investigations highlight its potential applications across diverse domains, emphasizing its impact on applied machine learning in various scientific fields.

Introduction

Automated code generation from natural language, a field often referred to as natural language programming (NLP), holds the promise of simplifying programming tasks and enhancing the software development process (Lei et al., 2013; Desai et al., 2016; Wang et al., 2023), particularly in the field of machine learning (ML) (Chandel et al., 2022). The demand for efficient ML solutions is continuously rising, showcasing the significance of this technology in streamlining programming tasks and enhancing software development processes. ML has transformed human lives and significantly impacted scientific research and engineering (Alpaydin, 2021). It has emerged as a standard tool in various domains, revolutionizing the way tasks are approached and problems are solved (Jung, 2022). With the increasing reliance on ML solutions, the ability to swiftly and accurately translate ambiguous task descriptions into functional code has become increasingly vital.

Early endeavors in code generation from natural language primarily rely on rule-based systems and template-based approaches (Gulwani, Polozov & Singh, 2017). These methods suffer from limited expressiveness and scalability as they struggle to accommodate the variability and complexity of human and coding languages (Allamanis et al., 2018).

Vaswani et al. (2017) introduce the Transformer architecture, a cornerstone in many natural language processing tasks, including code generation. Transformer-based models excel in capturing long-range dependencies and contextual information, leading to significant improvements in code generation quality. The synergy of deep learning techniques and the availability of extensive training data has transformed the landscape of code generation from natural language (Vaithilingam, Zhang & Glassman, 2022), paving the way for the development of large language models (LLMs). These LLMs exhibit the capability to learn intricate mappings between textual inputs and executable code.

While significant progress has been made in code generation from natural language, there remains a substantial gap in effectively transforming complex machine learning task descriptions into precise, executable code (Yin et al., 2022; Wen et al., 2024). Current generative models often produce common yet suboptimal code snippets based on textual input, failing to capture the nuanced requirements of specific ML tasks. This gap exists primarily due to the complexity and variability of ML tasks, which often require domain-specific knowledge and customized approaches. The challenge also lies in converting detailed ML task narratives into a structured series of code components, as LLMs excel more with direct instructions. By “instructions” we mean the high-level guidance provided to the model for generating specific outputs (see Methodology Section). Moreover, the difficulty is in maintaining coherence and logical flow throughout longer code sequences necessary for complete ML solutions. Addressing this knowledge gap can accelerate the development and prototyping of ML solutions, democratize ML development, and enhance the reproducibility and standardization of ML research.

Our approach, Linguacodus, seeks a more accurate and flexible solution. It involves a two-step process: first, it transforms the human-provided ML task descriptions into explicit, high-level instructions. This step ensures the instructions are clear, verifiable, and understandable to the user, laying a solid foundation for the next phase. Then, these high-level instructions are translated into machine-compilable code representation, specifically Python code in our case, with the potential for extension to other programming languages (Fig. 1). This method not only accommodates the intricate nature of ML tasks but also enhances the control and precision in the code generation process, meeting the need for understanding and controlled production of code in ML applications.

Figure 1 Linguacodus takes in the user-provided description of a machine learning task and generates an optimal solution instruction.

This instruction is then optionally refined using Multi-role LLM. Another LLM is employed to infer executable ML code based on the enhanced instruction. The resulting code represents the most effective solution for the specified task.

By converting human language into executable code, Linguacodus enables quick prototyping, ease iteration, and facilitate the deployment of ML models, potentially democratizing software development. This breakthrough allows individuals without extensive coding skills to engage in creating complex ML tasks, promoting innovation across various disciplines. The drive for such technology underlines a vision to broaden ML’s reach and impact, simplifying the development process and inviting a wider audience to contribute to technological advancements. Portions of this text were previously published as part of a preprint (Trofimova, Sataev & Ustyuzhanin, 2024).

Our main contributions can be summarized as follows:

A controllable transformation framework: We present a framework for the controlled transformation of ML task descriptions into solution instructions, involving fine-tuning the Llama 2 model using pairs of ML task descriptions and instructions.

Instruction-based sequential generation: We demonstrate the efficacy of executing instructions for sequential generation, producing compilable code with promising results based on evaluation metrics.

The rest of the article is organised as follows. ‘Related Work’ explores the application of LLMs in code generation, text-to-code conversion, controllable generation, and automating problem-solving tasks, shedding light on the limitations of LLMs in ML code synthesis. ‘Methodology’ provides an overview of the Linguacodus framework. ‘Experimantal Results and Analysis’ describes the experiments and validation of the approach, highlighting the effectiveness of Linguacodus in transforming plain English descriptions of ML tasks into executable code. ‘Discussion’ and ‘Limitations’ discusses and critically examines the limitations our approach. ‘Future Work’ suggest the future perspectives of the work. Finally, ‘Conclusion’ summarizes and concludes the article.

Related work

Code generation from developer’s requirements has emerged as a compelling area of research, bridging the realms of NLP and programming languages (Liu et al., 2020). Traditional methodologies for code synthesis from human language have historically leaned on formal semantic representations of natural language (Winograd, 1972; Harel et al., 1990; Buse & Weimer, 2012). However, formal specifications require manual creation and maintenance, making them labor-intensive and difficult to scale for large codebases or complex systems (Raychev, Vechev & Yahav, 2014).

Ling et al. (2016) automatically predict code snippets directly from natural language inputs by proposing latent predictor networks (LPN). LPN encapsulates the latent variable model for capturing the underlying structure of the input natural descriptions, and the predictor network for mapping the latent representations to corresponding code snippets.

Meanwhile, Rabinovich, Stern & Klein (2017), Yin & Neubig (2017) and Yin & Neubig (2018) emphasize the importance of incorporating syntax awareness into the neural network architectures. The researchers leverage the Abstract Syntax Tree to capture the well-defined structure in the target programming syntax. Additionally, long short-term memory (LSTM) networks are employed to capture long dependencies in natural language sequences. However, these methods predominantly rely on a single NL statement.

In contrast, Agashe, Iyer & Zettlemoyer (2019) tackle the task of interactive general-purpose code generation by incorporating a full sequence of previous natural language and code blocks as context within a Python Jupyter notebook environment (Kluyver et al., 2016). Still, the work is limited to the domain defined by the JuICe dataset, consisting of code snippets and corresponding markdowns, and does not utilize general task descriptions as inputs for code generation.

Utilizing vast amounts of code and natural language data has been made possible through pre-training techniques (Radford et al., 2018; Devlin et al., 2018). By leveraging pre-trained models, like CodeBERT (Feng et al., 2020), researchers strive to capture comprehensive representations of both code and language semantics. This enables the models to produce code from natural language descriptions that are not only more accurate but also contextually relevant. Such models offer versatility in code-related tasks, including code generation, summarization, and recommendation.

CoditT5 (Zhang et al., 2022) is another language model that generates edit-based output sequences from corrupted input sequences. Models like CoditT5 enhance code generation capabilities, aligning them more closely with user requirements.

Modern code generation approaches often rely on general-purpose transformers, exemplified by GPT-3. Codex (Chen et al., 2021), a notable model in this category, showcases the potential to generate code snippets directly from natural language prompts. AlphaCode (Li et al., 2022) extends this foundation, emphasizing the significance of code diversity and improving contextual understanding in LLMs.

In parallel, text-to-code conversion has gained prominence. PaLM-Coder (Chowdhery et al., 2023) presents a method for converting natural language descriptions into code, focusing on Java code generation. OpenAI models (Achiam et al., 2023; Bubeck et al., 2023) have further extended the capabilities of LLMs in understanding and generating code from textual prompts.

Controllable code generation is an emerging subfield with significant potential. CTRL (Keskar et al., 2019) is a conditional language model for controlled code generation. The model focuses on allowing users to specify conditions that influence the generated code, providing a level of control over the output. Texygen (Zhu et al., 2018) is a benchmarking platform for evaluating text generation models, including those designed for code generation. This platform facilitates the assessment of controllable code generation models by offering standardized evaluation metrics and tasks.

In automating problem-solving tasks, researchers have actively explored solutions such as AutoGluonTabular (Erickson et al., 2020) and H2O AutoML (LeDell & Poirier, 2020). These frameworks offer automated machine learning capabilities to streamline the model development process and improve prediction accuracy.

In particular, LightAutoML (Vakhrushev et al., 2021) tailors itself to the distinctive needs of large financial services companies companies. It provides solutions for handling large datasets with diverse types, non-stationary data, and specific validations, making it well-suited for complex financial analysis tasks.

Another recent AutoML framework, HuggingGPT (Shen et al., 2024), utilizes ChatGPT for task planning, model selection, subtask execution, and result summarization. HuggingGPT demonstrates versatility across a wide range of AI tasks, including natural language understanding and automated problem-solving.

Nair et al. (2023) present the dialog-enabled resolving agents (DERA), aiming for accurate output generation. DERA enhances the conversational abilities of LLMs by incorporating two distinct agent types: a researcher, responsible for processing information and identifying critical problem components, and a decider, capable of autonomously integrating the researcher’s information and making judgments on the final output. Although the DERA paradigm was initially used in healthcare, one can notice the potential applicability of multi-agent LLM in various training fields.

While automated machine learning offers structured workflow optimization, ML code generation based on natural language descriptions provides seamless integration into existing systems and customization for domain-specific tasks.

The recent advancements in code generation driven by LLMs have witnessed notable progress. Thus, OpenAI GPT models (Achiam et al., 2023; Bubeck et al., 2023), although not explicitly designed for code generation, have demonstrated proficiency in generating code snippets and understanding programming-related prompts. The generative capabilities of GPT models make them versatile tools for interpreting and translating natural language descriptions into executable code.

Google’s PaLM 2 (Anil et al., 2023) undergoes pre-training on a vast dataset encompassing web pages and source code, making it valuable for code debugging, completion, and generation across multiple programming languages. The model’s dual focus on semantic parsing and language model pre-training enhances its ability to comprehend and generate code based on diverse natural language inputs.

One of the leading publicly available LLMs for code generation is Code Llama (Roziere et al., 2023). An extension of Llama 2 (Touvron et al., 2023), Code Llama comes in two variations: a code producer and its instruction-specific refinement, Code Llama-Instruct. Code Llama-Instruct surpasses Code Llama in providing more helpful and secure responses in natural language, ensuring a more dependable performance. However, the generated instructions are generally broad-purpose and lack easy assessability regarding their suitability for specific tasks.

While OpenAI’s ChatGPT and similar LLMs have demonstrated remarkable capabilities in various natural language understanding tasks, they do have some inherent limitations in the context of ML code generation: 1) Lack of specificity: LLMs often generate code snippets that lack specificity for specific ML tasks. The generated code may be overly general and not finely tailored to the requirements of complex machine learning workflows.

2) Limited control over code generation: Users have limited control over the fine-tuning process of LLMs, making it challenging to enforce specific guidelines or constraints during the generation of ML code. This lack of control may result in variations in code quality and suitability for different tasks.

3) Handling ambiguity: Natural language descriptions of ML tasks can be inherently ambiguous. LLMs may struggle to disambiguate between multiple potential interpretations, leading to code snippets that may not accurately capture the intended meaning of the task.

4) Inability to learn task-specific patterns: While proficient in learning patterns from diverse data, LLMs may face challenges in capturing task-specific patterns relevant to ML code generation. This limitation can result in generated code that lacks the specificity required for specialized tasks.

5) Evaluation metrics and validation: The evaluation metrics for assessing the quality of generated code may not always align with the specific requirements of ML tasks. LLMs may prioritize generating syntactically correct code without necessarily ensuring the semantic correctness or optimization of the generated solutions.

Addressing these challenges requires a hybrid approach involving specialized ML code datasets and dimensional reduction within the learning space for LLM fine-tuning. The Code4ML (Drozdova et al., 2023) is a comprehensive corpus comprising of a) Kaggle challenge descriptions in natural language, b) Jupyter notebooks and their scores, c) Python code snippets, and d) competition-related metadata. This metadata includes formal descriptions of challenge datasets and scoring metrics. Code4ML relies on a knowledge taxonomy tree (Fig. 2) to categorize various Jupyter notebook code snippets. A description of a challenge solution in terms of the classes of this taxonomy significantly reduces the dimensionality of a code generation problem compared to the direct generation of code by using task description as a prompt. However, Code4ML lacks annotation for all code snippets. This limitation is addressed through the taxonomy-based categorization introduced by Berezovskiy et al. (2023).

Figure 2 Code4ML taxonomy tree.

Reproduced from Drozdova et al. (2023), with permission of the authors.

Methodology

This section presents a comprehensive overview of the Linguacodus. Figure 3 depicts the two stages of the framework. Initially, utilizing the fine-tuned Llama 2, we generate the most appropriate instruction, encapsulating the high-level core information of a generalized ML solution, tailored to a specific ML task. Subsequently, this instruction undergoes a sequential transformation into programming code through prompts with GPT-3.5.

Figure 3 Overall Linguacodus training framework.

Instruction creation

To extract the high-level code instructions, we’ve devised a four-stage framework: 1) High-level solution representation: We begin by creating high-level representations of ML solutions. To refine the quality of our dataset, the solutions undergo a ranking process based on their scores. Each solution is intricately linked to the natural language description of the ML task. Linguacodus utilizes the LLM to extract critical information regarding data preprocessing, model architecture, and the training procedure from existing code solutions. This information forms the high-level ML instruction. Figure 4 illustrates the precise input prompt presented to the model.

2) Llama 2 fine-tuning: Then, we utilize the acquired instructions as inputs for fine-tuning the open-source Llama 2 7b model. To ensure the relevance of the instructions to the machine learning (ML) task, we leverage the original code’s quality evaluation in the form of a score. The retrieved instructions are ranked based on their significance to the ML task. Furthermore, we furnish the Llama 2 model with essential information presented as prompts, including the task description, metric details, and data type information. The prompt-completion pair used in this stage is visually depicted in Fig. 5, with the separation marked by the [/INST] token. This comprehensive approach enhances the fine-tuning process, incorporating the quality ranking of instructions and pertinent task details for optimal model adaptation. Llama models have been pre-trained on vast amounts of data. By fine-tuning, we leverage this extensive knowledge and adapt it to specific tasks, often achieving state-of-the-art results with less data and time. The fine-tuning details are summarised in “Llama 2 Fine-Tunimg Details”.

3) Llama 2 inference: Next, we infer Llama 2 to select the top three most valuable instructions by specifying their rank using a dedicated prompt, as shown in Fig. 6.

4) Iterative enhancing LLM responses through multi-agent LLM: The inferred instructions then undergo further refinement with the assistance of multi-agent LLM. The primary goal of multi-agent LLM is to identify any logical errors in the provided instructions and subsequently choose the best option from the three variants, thereby enhancing the overall quality of the instructions. This intelligent processing is elucidated in Figs. 7 and 8.

Figure 4 Prompt for ML instructions retrieving.

Figure 5 Llama 2 fine-tune input.

Figure 6 Prompt for Llama 2 inference.

Figure 7 Prompt for multi-agent LLM for best instruction choice.

Figure 8 Prompt for multi-agent LLM for best instruction improvement.

ML code by instruction generation

The second stage of our approach centers on the actual code generation, building upon the instructions obtained in the previous step. In this phase, we harness the capabilities of language models to transform these instructions into functional and well-structured code that aligns with the underlying ML tasks.

Figure 9 precisely represents the sequential pipeline involved in the instruction-to-code transformation. We have separated the code synthesis into the stages of data preprocessing, model architecture, and model training. Additionally, we have also introduced a submission block to enable the testing of results on the Kaggle platform. The next step in this pipeline involves integrating all the generated code segments. To mitigate the possible execution problems, Linguacodus employs an error-fixing procedure, running it up to three times. In this process, the same LLM agent, responsible for integrating all code components iteratively, inputs the errors without any additional specifications.

Figure 9 Linguacodus instruction to code sequential transformation scheme.

This phase forms the critical bridge between the high-level ML instructions and the executable code, ensuring that the generated code adheres to the provided instructions and produces practical solutions for the intended ML tasks.

Experimental results and analysis

Dataset

Our research relies on the Code4ML dataset, focusing on Kaggle competitions encompassing all metric categories except ‘points,’ ‘significance,’ and ‘custom loss.’ We curate the top 75 solutions for retrieving high-level instructions from these competitions. It is essential to highlight that specific contests may have fewer than 75 solutions available for selection.

As a result, our training dataset comprises 395 natural language ML task descriptions paired with 7,023 corresponding Kaggle solutions. Figure 10 overviews the prevalent models featured in the selected solutions. Figure 11 illustrates the diversity of data types used in the chosen Kaggle competitions. This work emphasizes ML tasks involving tabular data. However, we do not restrict competitions to numeric tabular formats and consider those involving time series or text data presented with tables.

Figure 10 The most popular model choice among retrieved Kaggle solutions based on metric and data type.

Figure 11 The competitions distribution based on the data type.

To assess the effectiveness of our approach, we employ Kaggle competitions that are recent and popular, featuring more than 500 participating teams, ensuring that the tasks were unseen by our model. To approximate the distribution of the training competition space, we randomly select ten machine learning tasks, with a majority operating on numerical data and one each for text, time series, and image data.

Linguacodus generated instructions validation extends beyond the Kaggle platform, encompassing ML competitions hosted on CodaLab (Pavao et al., 2023). All the data used for validation and testing is not included in the training set.

Baseline

The overall comparative model for our framework is vanilla GPT-3.5, considering its prominence as a leading tool in natural language generation tasks. While other models exist, such as CodeBERT, CoditT5, PalM-Coder, and CTRL, their suitability for generating code from natural language task descriptions may be limited. Specifically, CodeBERT and CoditT5 are primarily trained for synthesizing code snippets rather than entire pipelines or comprehensive solutions. Therefore, GPT-3.5 is a more relevant and established benchmark in transforming natural language into complete machine learning pipelines. Additionally, GPT-3.5 demonstrates greater efficiency compared to Llama 2 (Zheng et al., 2024) and does not require payment, as GPT-4. Code Llama—Instruct is used as a reference model for the Linguacodus Instruction Creation phase.

Experiments setup and analysis

In our experiments, we use GPT-3.5 for retrieving instructions from the ML solutions, finding and improving the best instruction, and code generation. The selection of GPT-3.5 is driven by the consideration of balancing quality and inference time using the OpenAI API. However, the framework is generally agnostic to the choice of large language model, allowing for flexibility in utilizing different models based on specific requirements or preferences.

To underscore the significance of the research, we compare the instructions generated by the fine-tuned Llama 2 model and those inferred from Code Llama-Instruct. Our evaluation extends beyond the Kaggle platform, encompassing ML competitions hosted on CodaLab (Pavao et al., 2023) to ensure a thorough analysis. All the data used for validation and testing is not included in the training set. We use the selected by Linguacodus best instruction from the top three inferred by Llama 2. Additionally, we include examples of instructions automatically improved with the multi-agent LLM technique through the proposition of more advanced models for training.

Instructions produced by Code Llama-Instruct generally focus on the high-level approach and conceptual steps involved in training a model. They emphasize data preprocessing, model architecture, and training goals without delving into specific implementation details. In contrast, the fine-tuned Llama 2 instructions provide detailed, step-by-step breakdowns of the data preprocessing, model architecture, and model training processes. While the former offers a broader overview suitable for understanding the overall flow of the task, the latter caters to individuals seeking a more granular understanding, providing a comprehensive guide with specific library references and functions used at each stage (see “Sample Instructions Inferred by Code llama-Instruct and Fine-Tuned llama 2”).

Generating complete and functional code solutions using LLM requires providing the model with a detailed prompt or context outlining the task or problem statement. Hence, well-suited task instructions are vital for code generation. Our pipeline, enhanced by multi-agent LLM, can synthesize code via instructions of predefined quality, making our approach unique and promising for assisting in ML code generation. “Sample Code Genera by GPT-3.5 using Task Descriptions and Oure Refined Instructions” presents sample code generated by vanilla GPT-3.5 with automatically improved instructions and plain task descriptions. Raw GPT-3.5 output often contains code that cannot be compiled without further specific model training, whereas Linguacodus produces ready-to-run code.

Comparative results

Table 1 reports the Kaggle scores and percentiles obtained for code generated by Linguacodus and vanilla GPT-3.5 across a selection of randomly chosen machine learning tasks. Table D.1 provides an overview of the mapping between task IDs and corresponding Kaggle competition names. The percentiles reported in Table 1 reflect the relative standing on the Kaggle competition leaderboards, where lower percentiles indicate superior performance. The 0 percentile represents the top ranking, while higher percentiles indicate lower positions on the leaderboard. This comparison provides insight into how the generated solutions perform relative to the broader Kaggle community for each specific competition.

Table 1 The sample results of generated ML code validated on the Kaggle platform.

For each competition ID, the comparative scores and percentiles on the Kaggle competition leaderboard are provided. Lower percentiles indicate superior performance, × denotes an uncompilable solution. The best score results are highlighted in bold.

Id	Data type	Metric	Linguacodus score	Linguacodus percentile	GPT-3.5 score	GPT-3.5 percentile	
C1	Tabular	rmse	0.059	0	×	×	
C2	Tabular	roc-auc	0.948	66	0.943	68	
C3	Time series	rmse	15.409	0	24.136	21	
C4	Text	mcrmse	0.470	58	0.548	81	
C5	Tabular	roc-auc	0.773	77	0.752	80	
C6	Image	mean cosine similarity	0.714	0	×	×	
C7	Tabular	rmse	0.578	69	0.600	80	
C8	Tabular	mae	1.387	52	1.978	96	
C9	Tabular	mae	366.892	82	380.284	93	
C10	Tabular	roc-auc	0.862	76	×	×	

The use of Kaggle leaderboard percentiles provides a comprehensive assessment of the generated models. Unlike traditional code evaluation metrics, such as comparing Abstract Syntax Trees (Knuth, 1968) or using code similarity measures (Song et al., 2024), ML task performance requires a more nuanced approach. This is because the goal is to find the most effective solution for a given ML task, which can vary significantly in implementation while achieving similar results. Optimal solutions often emerge from novel combinations of existing ML techniques, making direct code comparison less relevant. Moreover, the effectiveness of generated code can only be truly measured by its performance on the specific ML task.

As shown in Table 1, Linguacodus consistently produces compilable code, outperforming vanilla GPT-3.5 solutions across specified machine learning metrics. Both Linguacodus and vanilla GPT-3.5 receive natural language descriptions and necessary metadata for each machine learning task as input. To ensure a fair and unbiased comparison, the code generated by both approaches undergoes up to three iterations of error treatment.

Kaggle, as a competitive platform, traditionally demands significant investment of time and expertise from its participants. Engaging in Kaggle competitions often requires deep understanding of the field and substantial time commitment. Our pipeline for transforming ML task descriptions into code offers a markedly more efficient alternative.

This approach significantly reduces the time and expertise required to bridge the gap between task descriptions and executable code, making machine learning development more accessible. While the OpenAI GPT-3.5 API generates a default solution (without error treatment process) in approximately 6 s, our pipeline averages 44 s on an A100 GPU. This process involves generating three instructions, correcting them, and sequentially generating code. Despite the longer processing time compared to GPT-3.5, our approach consistently yields superior results.

Discussion

As mentioned in ‘Related Work’, the recent advancements in code generation driven by LLMs have made significant strides, yet several challenges remain. Table 2 discusses how these issues are addressed with Linguacodus.

Table 2 Comparison of Linguacodus with other language models.

Issue	Description	
Lack of specificity	Linguacodus aims to provide more specific and tailored instructions for ML tasks by focusing on high-level information extraction rather than detailed code snippet classification in comparison with LLMs like CodeBERT (Feng et al., 2020) and CoditT5 (Zhang et al., 2022).	
Limited control over code generation	Ranked instructions allow for a controlled transformation process, providing a structured framework for code generation. While code-related (Anil et al., 2023; Roziere et al., 2023) and general-purpose (Achiam et al., 2023; Bubeck et al., 2023) LLMs do not offer the generation control tools, Linguacodus’ users can choose from the top-ranked instructions, offering control over the generated code.	
Handling ambiguity	The ranking process, coupled with the fine-tuning of LLMs, enhances the precision of instructions by prioritizing those that align most closely with the task descriptions, mitigating potential ambiguities, making Linguacodus on par or even better than OpenAI models (Bubeck et al., 2023), but open-source.	
Inability to learn task-specific patterns	Leveraging the fine-tuning process with Llama 2 7b on task-specific details allows the model to adapt and learn patterns specific to ML tasks, enhancing the quality and relevance of the generated instructions.	
Evaluation metrics and validation	Compared to evaluation metrics in models such as Texygen (Zhu et al., 2018), the ranking process, involving evaluation scores and task-specific details, is a robust validation mechanism for the generated instructions, ensuring their alignment with ML tasks and promoting solutions that adhere to evaluation metrics.	

Limitations

Despite the advancements presented by Linguacodus in addressing the challenges outlined in the section ‘Related Work’, there are several limitations that warrant consideration. The Code4ML dataset used to train Llama 2, which forms the foundation of Linguacodus, includes competitions only up to 2021. This temporal limitation means that the model may not fully cover the entire range of ML tasks and techniques, particularly recent emergent methods, potentially affecting its performance on cutting-edge problems.

Multi-agent LLM occasionally exhibits suboptimal performance compared to unprocessed Linguacodus instructions, emphasizing the role of context in task’s complexity. Ethical considerations surrounding biases and potential misuse of generated code highlight the need for responsible deployment. Linguacodus faces challenges when tasks deviate significantly from those fine-tuned on Llama 2, suggesting a need for dataset enrichment.

Insufficiently detailed instructions arise when tasks lack comprehensive descriptions, calling for more explicit task information. Recognizing that multi-agent LLM may not consistently outperform initially inferred instructions, human intervention is proposed to select the best instruction. This highlights the need for a balanced approach that combines the strengths of automated models with human judgment in refining outputs.

Future work

The temporal limitation of the training dataset underscores the importance of ongoing model updates and the potential for performance gaps in very recent or rapidly evolving areas of machine learning. This observation points to a development of a dynamic framework for enriching the ML data corpus. Such a framework would allow for continuous integration of new ML techniques, datasets, and competition results, ensuring that models like Linguacodus remain current and effective across the evolving landscape of machine learning tasks.

Another promising direction for future work involves exploring alternative, more deterministic approaches to constructing high-level instructions. One such approach is the development of a graph-instruction methodology. This could enable a more structured representation of the ML task, allowing for better assessment of intermediate generation steps and interpretability. By mapping the natural task description to a graph-based representation, we could potentially achieve greater transparency in the instruction generation process, facilitating easier evaluation and refinement of the model’s outputs.

Conclusion

In this article, we introduce a comprehensive approach to transforming unstructured ML task descriptions into executable code, presenting the novel Linguacodus model. Leveraging the Code4ML dataset, which encompasses a rich collection of Python code snippets, contest summaries, and data descriptions from Kaggle competitions, our methodology capitalizes on the dataset’s valuable competition-related metadata, data types, and scoring metrics. Inspired by the knowledge taxonomy tree introduced in Drozdova et al. (2023), we adopt a similar organizational framework to achieve dimensional reduction in our ML task description-to-code synthesis approach. However, our approach differs in that it focuses on high-level information extraction rather than individual code snippet classification. This strategic shift simplifies and streamlines the code generation process, making it more efficient and adaptable.

Linguacodus is structured into two phases: synthesizing high-level ML solution instructions and transforming these instructions into functional code. To generate instructions, the Llama 2 model is fine-tuned on the Code4ML corpus. The top three instructions are then inferred and further refined with the assistance of multi-agent LLM, ensuring the highest quality instructions for subsequent code generation. The second phase involves translating these refined instructions into well-structured and executable code segments, encompassing data preprocessing, model architecture, model training, and submission block generation. This transformation bridges the gap between high-level ML instructions and practical code, ensuring alignment with the underlying ML tasks.

Our approach’s effectiveness is validated through experiments on Kaggle competitions that are not part of our training data. The results demonstrate that the generated code is compilable and aligns well with the evaluation metrics. We also compare the performance of multi-agent LLM and unprocessed Code Llama—Instructions, highlighting the need for further refinement in multi-agent LLM’s algorithmic approach to achieve superior solution quality consistently.

In summary, the research provides an innovative and efficient solution for code generation from ML task descriptions, showcasing the capabilities of Linguacodus. By capitalizing on the Code4ML dataset’s wealth of resources and introducing a structured approach to instruction synthesis and code generation, we bridge the gap between natural language task descriptions and executable code, making machine learning development more accessible and efficient.

llama 2 fine-tuning details

To align natural language descriptions of machine learning tasks with high-level code instructions extracted from ML code solutions, we fine-tune the Llama 2 model. Table A.1 presents the hyperparameters used in the Llama 2 fine-tuning process.

Table A.1 Llama 2 fine-tuning hyper-parameters.

LoRA parameters	
LoRA attention dimension	64	
Alpha parameter for LoRA scaling	16	
Dropout probability for LoRA layers	0.1	
4-Bit precision parameters	
4-bit precision base model loading	True	
Compute dtype for 4-bit base models	float16	
Quantization type	nf4	
Nested quantization for 4-bit base models	False	
Training arguments parameters	
Number of training epochs	1	
Enable fp16/bf16 training	False/False	
Batch size per GPU for training	4	
Batch size per GPU for evaluation	4	
Enable gradient checkpointing	True	
Maximum gradient normal	0.3	
Initial learning rate	2e-4	
Weight decay	0.001	
Optimizer	AdamW	
Learning rate schedule	constant	
Number of training steps	−1	
Ratio of steps for a linear warmup	0.03	
Group sequences into same length batches	True	
Save checkpoint every X updates steps	500	
Log every X updates steps	25	
Sequence Fine-Tuning Parameters	
Maximum sequence length	None	

sample instructions inferred by code llama—instruct and fine-tuned llama 2

This section presents a comparative analysis of instructions for various machine learning tasks generated by three methods: Code Llama—Instruct; fine-tuned Llama 2 (best instructions selected by Linguacodus); multi-agent LLM automatic improvement.

Our analysis focuses on four competitions: two from CodaLab and two from Kaggle. Table B.1 summarizes the key information for these selected competitions. The set of tasks represented in these competitions allows for a comprehensive comparison. We use the competition names and task descriptions as prompts for instruction generation.

Table B.1 Competitions information.

Competition name	Source	Data type	Metric type	Task description	
“SHROOM—a Shared-task on Hallucinations and Related Observable Overgeneration Mistake” (Pavao et al., 2023)	CodaLab	Text	Accuracy	SHROOM challenges participants to identify outputs that are grammatically correct but semantically inaccurate, in a binary classification task. The competition focuses on detecting “hallucinations” in outputs from neural language models across definition modeling, machine translation, and paraphrase generation, with or without access to the model that generated these outputs	
“Climate Activism Stance and Hate Event Detection Shared Task” (Pavao et al., 2023)	CodaLab	Text	f1 score	The CASE 2024 competition at EACL 2024 focuses on three subtasks in the context of climate activism: identifying hate speech, detecting targets of hate speech (individuals, organizations, communities), and determining the stance (support, oppose, neutral) of given texts.	
“Stable diffusion—image to prompts” (Reade, Cukierski & Chow, 2023)	Kaggle	Images	Mean cosine similarity	The competition aims to develop a model that can predict the text prompt that corresponds to a given generated image, challenging the understanding of the relationship between text prompts and images in text-to-image models. The goal is to create embeddings for predicted prompts, ensuring robust prompt similarity assessment, and exploring prompt engineering in the context of image generation.	
“Regression with a tabular Paris housing price dataset” (Reade & Chow, 2023f)	Kaggle	Tabular	Rmse	Regression with a Tabular Paris Housing Price Dataset	

Tables B.2–B.4 showcase the retrieved instructions for CodaLab competition “SHROOM—a Shared-task on Hallucinations and Related Observable Overgeneration Mistake”. The first instruction adopts a high-level approach, focusing on the overarching strategy and conceptual steps involved in training a model to identify “hallucinations” in neural language model outputs. It significantly emphasizes data preprocessing, model architecture, and training objectives while avoiding intricate implementation details.

Table B.2 A sample instruction generated for “SHROOM—a Shared-task on Hallucinations and Related Observable Overgeneration Mistake” competition by Code Llama - Instruct.

As a data analyst, I would start by preprocessing the text data to ensure that it is in a format that can be easily analyzed and processed. This may involve removing stop words, stemming or lemmatizing the text, and converting the text to a numerical representation using techniques such as bag-of-words or word embeddings.	
Once the data is preprocessed, I would then build a binary classification model that can accurately detect grammatically correct but semantically inaccurate outputs. This model architecture could include a combination of techniques such as word embeddings, recurrent neural networks (RNNs), and convolutional neural networks (CNNs) to capture the nuances of language and detect patterns in the data.	
To train the model, I would use a dataset of labeled examples, where each example is annotated with a binary label indicating whether the output is grammatically correct but semantically inaccurate or not. I would then use this dataset to train the model using a suitable machine learning algorithm, such as logistic regression or support vector machines (SVMs), and evaluate its performance using metrics such as accuracy, precision, recall, and F1 score.	
Overall, the goal of this task is to develop a model that can accurately detect “hallucinations” in outputs from neural language models, and to evaluate its performance using appropriate metrics. By preprocessing the data and using a suitable model architecture, I believe that I can develop a model that can accurately detect these hallucinations and improve the quality of the outputs generated by the neural language models.	

Table B.3 The best out of top-3 sample instruction generated for “SHROOM—a Shared-task on Hallucinations and Related Observable Overgeneration Mistake” competition by fine-tuned Llama 2.

Data preprocessing:	
- The training data is loaded from a CSV file using pandas.	
- The test data is loaded from a CSV file using pandas.	
- The training data is split into training and validation sets using the train_test_split function from sklearn.	
- The training data is further processed by converting the text data into tokenized sequences using the Tokenizer class from nltk.	
- The tokenized sequences are then converted into input IDs and attention masks using the TensorFlow SequenceClassification API.	
Model architecture:	
- The model architecture used is a BERT-based model.	
- The BERT model is loaded from a pre-trained checkpoint using the BERTCheckpoint class from the TensorFlow BERT library.	
- The BERT model is then fine-tuned on the training data using the TensorFlow TensorBoard API.	
- The fine-tuned BERT model is used as the base model for the classification task.	
Model training:	
- The model is trained using the TensorFlow TensorBoard API.	
- The training data is used to train the model for a specified number of epochs.	
- The training progress is monitored using the TensorFlow TensorBoard API.	
- The trained model is then used to make predictions on the test data.	
- The predictions are saved in a submission file for evaluation.	

Table B.4 An automatically chosen and improved with multi-agent LLM best sample instruction generated for “SHROOM - a Shared-task on Hallucinations and Related Observable Overgeneration Mistake” competition by fine-tuned Llama 2.

Data preprocessing:	
∙ Load the training and test data from CSV files, ensuring that the data is properly formatted.	
∙ Preprocess the text data to prepare it for model input. This may include tokenization and data cleaning if necessary.	
Model architecture:	
∙ Choose an appropriate model architecture for this binary classification task. Consider using pre-trained models like BERT or GPT-3 as a starting point.	
∙ Fine-tune the selected model on the training data to adapt it to the specific task. Ensure that the architecture includes appropriate layers for text processing and classification.	
Model training:	
∙ Set up the training process with the following components:	
∘ Loss Function: Use binary cross-entropy as the loss function for binary classification.	
∘ Optimizer: Select an optimizer like Adam.	
∘ Metrics: Track accuracy as the evaluation metric.	
∙ Train the model for a specified number of epochs, monitoring the training progress to avoid overfitting.	
∙ Split the data into training and validation sets to evaluate the model’s performance during training.	
Model evaluation:	
∙ After training, evaluate the model on the validation set to assess its accuracy and other relevant metrics.	
∙ Make necessary adjustments to the model or training process based on the evaluation results.	
Model evaluation:	
∙ Use the trained and validated model to make predictions on the test data.	
∙ Save the predictions in a submission file for evaluation.	

In contrast, the subsequent instruction provides a meticulous, step-by-step breakdown of the data preprocessing, model architecture, and model training procedures. It intricately outlines actions such as data loading using pandas, tokenization with the specific Tokenizer class, fine-tuning a BERT-based model through the TensorFlow BERT library, and tracking training progress using the TensorFlow TensorBoard API. This level of detail makes it more suitable for direct implementation.

An instruction automatically improved with multi-agent LLM goes beyond by suggesting a more advanced model for solving an ML task. Additionally, it specifies the optimization algorithm and training metric, providing a more refined and advanced set of instructions for users. Thus, the potential of multi-agent LLM in enhancing the quality and sophistication of instructions for machine learning tasks.

Tables B.5–B.13 depict the triple instructions for the competitions “Climate Activism Stance and Hate Event Detection Shared Task”, “Stable Diffusion—Image to Prompts” and “Regression with a Tabular Paris Housing Price Dataset”, respectively. Once again, the fine-tuned Llama 2 yields precise and straightforward instructions for generating accurate code. Notably, these instructions are the optimal choice for the assigned tasks, aligning seamlessly with the prompt specifications and achieving the highest solution rating.

Table B.5 A sample instruction generated for “Climate Activism Stance and Hate Event Detection Shared Task” competition by Code Llama—Instruct.

As a data analyst, I would start by preprocessing the data to ensure that it is in a format that can be easily analyzed and processed. This may involve cleaning the data, removing any irrelevant or redundant information, and transforming the data into a format that can be easily analyzed by a machine learning model.	
Once the data is preprocessed, I would then move on to designing the model architecture. For the task of identifying hate speech, detecting targets of hate speech, and determining the stance of given texts, a combination of natural language processing (NLP) and machine learning techniques would be necessary.	
To identify hate speech, I would use a technique called sentiment analysis, which involves analyzing the sentiment of a piece of text to determine whether it is positive, negative, or neutral. To detect targets of hate speech, I would use a technique called entity recognition, which involves identifying and extracting specific entities such as individuals, organizations, and communities from the text.	
To determine the stance of given texts, I would use a technique called stance detection, which involves analyzing the sentiment of a piece of text to determine whether it is supportive, opposing, or neutral.	
Once the model architecture is designed, I would then train the model using the preprocessed data. The training process would involve feeding the preprocessed data into the model and adjusting the model’s parameters to minimize the error between the model’s predictions and the true labels.	
Throughout the training process, I would monitor the model’s performance using metrics such as accuracy, precision, recall, and F1-score. Once the model is trained, I would evaluate its performance on a separate test set to ensure that it is generalizing well to new data.	
Overall, the key to success in this machine learning task is to carefully preprocess the data, design an appropriate model architecture, and train the model using a large and diverse dataset. By following these steps, I believe that it is possible to achieve high performance on the CASE 2024 competition at EACL 2024.	

Table B.6 A sample instruction generated for “Climate Activism Stance and Hate Event Detection Shared Task” competition by Code Llama-Instruct.

Data preprocessing:	
- Read the training and test data from CSV files.	
- Perform data cleaning and preprocessing steps, such as handling missing values, converting categorical variables to numerical codes, and encoding text data.	
- Split the data into training and testing sets.	
Model architecture:	
- Define the model architecture using a pre-trained language model (e.g., BERT) and a classification head.	
- The model will be trained using a combination of binary cross-entropy loss and categorical cross-entropy loss.	
Model training:	
- Use the training data to train the model.	
- Use the validation data to evaluate the model’s performance.	
- Use the testing data to make predictions	

Table B.7 An automatically chosen and improved with multi-agent LLM best sample instruction generated for “Climate Activism Stance and Hate Event Detection Shared Task” competition by fine-tuned Llama 2.

Data preprocessing:	
∙ Clean and normalize the data, removing irrelevant or redundant information.	
∙ Handle missing values and outliers appropriately.	
Model architecture:	
∙ Select a suitable machine learning algorithm for text classification based on the task’s requirements and data characteristics. Consider using natural language processing (NLP) techniques.	
Data splitting:	
∙ Split the dataset into training, validation, and test sets to ensure proper model evaluation.	
Model training and hyperparameter tuning:	
∙ Train the model on the training dataset, adjusting parameters to minimize error. Perform hyperparameter tuning to optimize model performance.	
Evaluation:	
∙ Calculate the F1-score to evaluate the model’s performance, ensuring it aligns with the competition’s metric type	
Handling imbalanced data:	
∙ Implement techniques to address imbalanced class distribution if necessary.	

Table B.8 A sample instruction generated for “Stable Diffusion-Image to Prompts” competition by Code Llama-Instruct.

As a data analyst, I would start by preprocessing the image data to ensure that it is in a format that can be used for machine learning. This may involve resizing the images, normalizing the pixel values, and possibly applying data augmentation techniques to increase the size of the dataset.	
Once the data is preprocessed, I would then design a model architecture that is suitable for the task at hand. This may involve using a combination of convolutional neural networks (CNNs) and recurrent neural networks (RNNs) to capture the relationships between text prompts and images.	
To train the model, I would use a mean cosine similarity metric to measure the similarity between the predicted prompts and the actual prompts. I would also use a technique called prompt engineering to ensure that the predicted prompts are robust and accurate. This involves using a combination of techniques such as word embedding, attention mechanisms, and recurrent neural networks to improve the performance of the model.	
Overall, the goal of this task is to develop a model that can predict the text prompt that corresponds to a given generated image, and to explore the relationship between text prompts and images in the context of text-to-image models. By using a combination of data preprocessing, model architecture, and model training, I believe that we can achieve this goal and make significant contributions to the field of natural language processing and computer vision.	

Table B.9 The best out of top-3 sample instruction generated for “Stable Diffusion-Image to Prompts” competition by fine-tuned Llama 2.

Data preprocessing:	
∙ The training data consists of 100,000 images and their corresponding text prompts. The test data consists of 10,000 images and their corresponding text prompts.	
∙ The images are resized to a fixed size of 224 × 224 pixels. ∙ The text prompts are tokenized using the tokenizer from the BERT model.	
∙ The tokenized text prompts are converted to integers using the tokenizer’s convert_tokens_to_ids method.	
∙ The image IDs and text prompt IDs are stored in separate arrays.	
Model architecture:	
∙ The model architecture used is a BERT-based model.	
∙ The BERT model is loaded from a pre-trained checkpoint.	
∙ The BERT model is used as a feature extractor.	
∙ The extracted features are passed through a linear layer with a sigmoid activation function.	
∙ The sigmoid activation function is used to convert the output of the linear layer to a probability value.	
Model training:	
∙ The model is trained using the training data.	
∙ The training data is split into training and validation sets.	
∙ The model is trained using the Adam optimizer and a learning rate of 1e-5. The training process is repeated for a specified number of epochs.	
∙ The model is evaluated on the validation set during training.	
∙ The training and validation loss and accuracy are recorded.	
Additional information:	
∙ The model is saved after training.	
∙ The saved model is loaded for use in the test data.	
∙ The test data is passed through the model to generate predictions.	
∙ The predicted text prompts are stored in a list.	
∙ The predicted text prompts are then converted to integers using the tokenizer’s convert_tokens_to_ids method.	
∙ The predicted text prompts are saved in a submission file.	

Table B.10 An automatically chosen and improved with multi-agent LLM best sample instruction generated for “Stable Diffusion—Image to Prompts” competition by fine-tuned Llama 2.

Data preprocessing:	
∙ Load the training and test data, ensuring that both image and text prompt data are correctly loaded and aligned.	
∙ Apply any necessary data preprocessing, including handling missing values, cleaning the data, and addressing class imbalances if present.	
∙ Perform image preprocessing, such as resizing and normalizing pixel values.	
∙ Tokenize the text prompts using a suitable tokenizer, considering any specific requirements for this task.	
∙ Convert tokenized text prompts into numerical representations (embeddings) using a method that captures the semantic meaning of the text effectively.	
Model architecture:	
∙ Design a model architecture that combines image and text information effectively. Consider using a combination of convolutional neural networks (CNNs) for image data and recurrent neural networks (RNNs) for text data.	
∙ Implement a mechanism for merging or concatenating the image and text features within the model architecture.	
∙ Utilize pre-trained embeddings or models if available and relevant to improve model performance.	
Model training and evaluation:	
∙ Split the training data into training and validation sets for model training. Implement a suitable validation strategy, such as k-fold cross-validation, to assess model performance robustly.	
∙ Train the model with appropriate hyperparameters, including learning rate, batch size, and optimizer choice. Monitor training with metrics relevant to the task, such as mean cosine similarity.	
∙ Implement early stopping to prevent overfitting and save the best-performing model during training.	
∙ Evaluate the model using metrics like mean cosine similarity on the validation set to assess its performance.	
∙ Consider strategies like learning rate schedules or fine-tuning pre-trained models to improve convergence and model generalization.	

Table B.11 A sample instruction generated for “Regression with a Tabular Paris Housing Price Dataset” competition by Code Llama-Instruct.

As a data analyst, I would follow these steps to solve the regression problem with the tabular Paris housing price dataset:	
Data preprocessing:	
Check for missing values and handle them appropriately.Scale the data using a standard scaler or normalizer to ensure that all features are on the same scale.Split the data into training and testing sets	
Model architecture:	
Use a linear regression model with a single hidden layer to predict the housing prices.The model should have an input layer with the number of features in the dataset, a hidden layer with a suitable number of neurons, and an output layer with a single neuron.The model should also have an activation function for the hidden layer, such as ReLU or tanh	
Model training:	
Train the model using the training set and evaluate its performance on the testing set using the RMSE metric.Adjust the hyperparameters of the model, such as the learning rate, number of hidden neurons, and regularization strength, to improve the model’s performance.Repeat the training and evaluation process until the model’s performance on the testing set is satisfactory.	
By following these steps, I can help solve the regression problem with the tabular Paris housing price dataset using a linear regression model with a single hidden layer.	

Table B.12 The best out of top-3 sample instruction generated for “Regression with a Tabular Paris Housing Price Dataset” competition by fine-tuned Llama 2.

Data preprocessing:	
-Read the training and testing data from CSV files.	
-Check for missing values in the dataset and fill them with appropriate values. Perform exploratory data analysis to understand the distribution of variables and any correlations.	
-Transform categorical variables into numerical codes.	
-Scale numerical variables using standard scaling techniques.	
Model architecture:	
-Define the model architecture using a suitable regression algorithm (e.g., Linear Regression, Ridge Regression, Lasso Regression, Elastic Net Regression).	
-Define the hyperparameters for the model (e.g., regularization strength, learning rate).	
Model training:	
-Split the data into training and testing sets.	
-Train the model using the training data.	
-Evaluate the model using the testing data.	
-Fine-tune the model by adjusting the hyperparameters.	
Model prediction:	
-Use the trained model to predict the target variable for the testing data.	
-Calculate the root mean squared error (RMSE) between the predicted values and the actual values.	
-Save the predicted values in a CSV file for submission.	
Note: The code provided is just an example and may need to be modified based on the specific requirements of the task	

Table B.13 An automatically chosen and improved with multi-agent LLM best sample instruction generated for “Regression with a Tabular Paris Housing Price Dataset” competition by fine-tuned Llama 2.

Data preprocessing:	
∙ Read the training and testing data from CSV files.	
∙ Check for missing values in the dataset and fill them with appropriate values.	
∙ Perform exploratory data analysis to understand the distribution of variables and any correlations.	
∙ Handle categorical variables by transforming them into numerical codes or using one-hot encoding.	
∙ Scale numerical variables using standard scaling techniques.	
Model selection and training:	
∙ Choose an appropriate regression algorithm considering the dataset’s characteristics. You may consider Linear Regression, Ridge Regression, Lasso Regression, or Elastic Net Regression. The choice should be based on the nature of the problem, linearity assumptions, and potential multicollinearity.	
∙ Define the hyperparameters for the selected regression model. Pay attention to regularization strength, learning rate, and any other relevant hyperparameters.	
∙ Split the data into training and testing sets for model evaluation. Additionally, consider using cross-validation for a more robust assessment.	
∙ Train the selected regression model using the training data. ∙ Validate the model assumptions such as linearity, homoscedasticity, and independence of errors.	
∙ Evaluate the model using the testing data. Calculate the root mean squared error (RMSE) to assess the model’s predictive performance.	
∙ Fine-tune the model by adjusting hyperparameters as needed to improve its performance.	

Multi-agent LLM enriches the instructions by delving deeper and justifying the ML steps. This enhancement contributes to the clarity of the instructions and adds a layer of transparency, aiding users in understanding the rationale behind the suggested approach. The collaborative synergy between fine-tuned Llama 2 and multi-agent LLM demonstrates the potential for leveraging advanced models to refine and augment machine-generated instructions.

sample code generated by gpt-3.5 using task descriptions and our refined instructions.

Tables C.1 and C.2 represent the examples of code inferred by GPT-3.5 with two variations of task-describing prompts: one with and one without the automatically chosen best instruction. Using a pure task description prompt may result in incomplete ML code generation, lacking an adequately defined model, for example. Conversely, when the task description prompt is enriched with the instruction, GPT-3.5 is driven to produce compilable results1 .

Table C.1 Code parts generated for Kaggle competition “Stable Diffusion-Image to Prompts” with and without an enriched instruction prompt.

Without instruction	With instruction	
import torch
import torchvision.transforms as transforms
from some_image_to_text_model import
     ImageToTextModel
from some_text_embedding_model import
     TextEmbeddingModel
import pandas as pd	# Import necessary libraries
***
import tensorflow as tf
from tensorflow.keras.preprocessing.image import
ImageDataGenerator
from tensorflow.keras.preprocessing.text import
Tokenizer	
# Load your pre-trained models
image_to_text_model = ImageToTextModel.
     load_pretrained(′
     path_to_image_to_text_model′)
text_embedding_model = TextEmbeddingModel.
     load_pretrained (′
     path_to_text_embedding_model′)
# Function to predict prompt from image
def predict_prompt(image_path):
   # Load and preprocess the image
   image = Image.open (image_path)
   transform = transforms.Compose ([
      transforms.Resize ((224, 224)),
      transforms.ToTensor(),
   ])
   image = transform(image).unsqueeze(0)
   # Use the model to predict the prompt
   predicted_prompt = image_to_text_model.
     predict (image)
   return predicted_prompt
# Function to convert prompt to embedding
def prompt_to_embedding(prompt):
   # Convert text prompt to embedding
   embedding = text_embedding_model.encode
      ( prompt)
   return embedding
***	from tensorflow.keras.layers import Input, Embedding,
      LSTM, Conv2D, Flatten, concatenate
from tensorflow.keras.models import Model
***
# Data Preprocessing
# Load training and test data
train_data = pd.read_csv (′train_data.csv′)
test_data = pd.read_csv(′test_data.csv′)
# Preprocess text prompts
tokenizer = Tokenizer()
tokenizer.fit_on_texts (train_data[′prompt′])
train_sequences = tokenizer.texts_to_sequences (
     train_data[′prompt′])
test_sequences = tokenizer.texts_to_sequences (test_data[
     ′prompt′])
# Pad sequences to a fixed length if necessary
max_seq_length = 20
train_sequences =tf.keras.preprocessing.sequence.
      pad_sequences (train_sequences, maxlen=max_seq_length)
test_sequences = tf.keras.preprocessing.sequence.
      pad_sequences (test_sequences, maxlen=max_seq_length)
# Image preprocessing (resize, normalize)
image_size = (128, 128)
train_datagen = ImageDataGenerator (rescale=1./255,
      validation_split = 0.2)
train_generator = train_datagen.flow_from_dataframe (***) valid_generator = train_datagen.flow_from_dataframe (***)
# Model Architecture
# Define text input
text_input = Input (shape = (max_seq_length,))
text_embedding = Embedding (input_dim = len (tokenizer.
     word_index) + 1, output_dim=128) (text_input)
text_lstm = LSTM(128) (text_embedding)
# Define image input
image_input = Input(shape = (image_size[0], image_size [1],
     3))
image_cnn = Conv2D(64, (3, 3), activation=′relu′)(
     image_input)
image_flatten = Flatten()(image_cnn)
***	

Table C.2 Code parts generated for Kaggle competition “Regression with a Tabular Paris Housing Price Dataset” with and without an enriched instruction prompt.

Without instruction	With instruction	
import pandas as pd
from sklearn.model_selection import
    train_test_split
from sklearn.ensemble import
    RandomForestRegressor
from sklearn.metrics import mean_squared_error
import numpy as np	import pandas as pd
import numpy as np
from sklearn.model_selection import train_test_split
from sklearn.preprocessing import StandardScaler
from sklearn.linear_model import LinearRegression, Ridge, Lasso, ElasticNet	
# Load the data
train_data = pd.read_csv (′train.csv′)
test_data = pd.read_csv(′test.csv′)
# Separate features and target
X = train_data.drop (′price′, axis=1)
y = train_data [′price′]
# Split the data into training and validation sets
X_train, X_val, y_train, y_val =
     train_test_split (X, y, test_size = 0.2,
     random_state=42)
# Initialize the model
model = RandomForestRegressor (random_state=42)
# Train the model
model.fit (X_train, y_train)
# Predict on validation set
val_predictions = model.predict (X_val)
# Calculate RMSE
rmse = np.sqrt(mean_squared_error (y_val,
     val_predictions))
print (f′Validation RMSE: rmse′)
# Predict on test data
test_predictions = model.predict (test_data)
***	from sklearn.metrics import mean_squared_error
# Data Preprocessing
# 1. Read the training and testing data from CSV files.
train_data = pd.read_csv (′train.csv′)
test_data = pd.read_csv (′test.csv′)
test_data = pd.read_csv (′test.csv′)
# 2. Check for missing values in the dataset and fill       them with appropriate values. train_data.fillna(train_data.mean(), inplace = True)
test_data.fillna (test_data.mean(), inplace=True)
***
# 5. Scale numerical variables using standard scaling techniques.
scaler = StandardScaler()
train_data [[′squareMeters′, ′numberOfRooms′, ′floors′,      ′numPrevOwners′, ′made′, ′garage′]] = \
   scaler.fit_transform (train_data [[′squareMeters′,       ′numberOfRooms′, ′floors′, ′numPrevOwners′, ′made′, ′garage′]])
test_data*** = \
   scaler.transform (***)
# Model Selection and Training
# 1. Choose an appropriate regression algorithm  considering the dataset′s characteristics.
# Let′s use Ridge Regression as an example.
regression_model = Ridge (alpha=1.0) # You can adjust alpha as needed.
# 3. Split the data into training and testing sets for model evaluation.
X = train_data.drop (columns=[′id′, ′price′])
y = train_data[′price′]
X_train, X_test, y_train, y_test = train_test_split (X, y
, test_size = 0.2, random_state = 42)
# 4. Train the selected regression model using the training data. regression_model.fit (X_train, y_train)
# 5. Evaluate the model using the testing data.
y_pred = regression_model.predict (X_test)
rmse = np.sqrt (mean_squared_error (y_test, y_pred))
print (f′Root Mean Squared Error (RMSE): {rmse}′)
# Model Prediction and Submission
# 1. Use the trained model to predict the target variable for the testing data.
test_features = test_data.drop (columns=[′id′])
predicted_prices = regression_model.predict (
;   test_features)
***	

list of the competitions used for validation.

This section provides a list of the competitions used to validate our approach. Table D.1 presents a mapping between competition IDs and their corresponding names. These competitions, sourced from the Kaggle platform, represent a range of machine learning tasks including regression, binary classification, and specialized problems like image-to-prompt generation.

Table D.1 Competition ID to competition name mapping.

Competition Id	Competition name	
C1	“Feature imputation with a heat flux dataset” (Reade & Chow, 2023a)	
C2	“Binary classification of machine failures” (Reade & Chow, 2023b)	
C3	Predict CO2 emissions in rwanda (Moruri et al., 2023)	
C4	CommonLit - evaluate student summaries (Franklin, 2023)	
C5	Binary classification with a tabular credit card fraud dataset (Reade & Chow, 2023c)	
C6	Stable diffusion-image to prompts (Reade, Cukierski & Chow, 2023)	
C7	Regression with a tabular california housing dataset (Reade & Chow, 2023d)	
C8	Regression with a crab age dataset (Reade & Chow, 2023e)	
C9	Regression with a wild blueberry yield dataset (Reade & Chow, 2023g)	
C10	Binary classification with a bank churn dataset (Reade & Chow, 2024)	

We would like to express our appreciation to Denis Derkach and Artem Maevskiy for their invaluable comments and support.

Additional Information and Declarations

Competing Interests

Author Contributions

Data Availability

1 Throughout this article, ‘***’ indicates code segments extracted by the authors.

The authors declare that they have no competing interests.

Ekaterina Trofimova conceived and designed the experiments, performed the experiments, analyzed the data, performed the computation work, prepared figures and/or tables, authored or reviewed drafts of the article, and approved the final draft.

Emil Sataev performed the experiments, analyzed the data, performed the computation work, prepared figures and/or tables, and approved the final draft.

Andrey Ustyuzhanin conceived and designed the experiments, authored or reviewed drafts of the article, and approved the final draft.

The following information was supplied regarding data availability:

The used dataset is open-access Code4ML corpus available at Zenodo: Anastasia Drozdova, Polina Guseva, Ekaterina Trofimova, Anna Scherbakova, Andrey Ustyuzhanin, Anastasia Gorodilova, & Valeriy Berezovskiy. (2022). Code4ML: a Large-scale Dataset of annotated Machine Learning Code (1.0.1) [Data set]. Zenodo. https://doi.org/10.5281/zenodo.7733823.

The code is available at Zenodo: Ekaterina Trofimova, & Deltax2016. (2024). ketrint/Linguacodus: v1.0.0 (v1.0.0). Zenodo. https://doi.org/10.5281/zenodo.13283827.

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
