# Peer review of "Linguacodus: a synergistic framework for transformative code generation in machine learning pipelines"

_PeerJ Computer Science, doi:10.7717/peerj-cs.2328_

## Round 0.1 · original submission · Minor Revisions

· Academic Editor

Minor Revisions

Please revise the article carefully according to the comments. Then it will be evaluated again.

·

Basic reporting

The authors should improve to improve overall organization of the manuscript.
Please arrange the figures in more professional way and not just copy/paste presentation format. Resolution of the figures should be improved. Try to use linker for Tables/Figures/References in the text to enhance readability. Try to use less paragraphs if not actually required, like say in "Conclusion" section. It impedes readability.
The authors discussed limitations of prior works in "Related work" section and again in a separate section as "Discussion" as a comparative analysis. While this is relevant but appears as dis-joint and repetitive. It will be more appropriate to present in the "Discussion" section only as a comparative analysis against cited previous research works [for each feature/points] in a tabular format (not mandatory), how the proposed approach outperforms or performing as per.
"Limitations" should be discussed as a separate section. What will be the future directives?

Experimental design

Research motivation, Proposed methodology and training criterion are well discussed and well presented.
Figures helped better understandability of each stages in "instruction creation" section.

Validity of the findings

The proof-of-validation is satisfactory to support proposed claims. However, authors need to provide additional details as the "score" metrics for proposed "Linguacodus" and that of GPT3.5, is the score "higher" is more accurate or the "lower", as from the percentile also, it appears GPT3.5 is outperforming the proposed approach, say for competition C3, "Linguacodus" percentile 0 against GPT3.5 21, for C4, the former 58 against the later 81 and so on. Please justify "Linguacodus consistently produces compilable code, outperforming vanilla 298 GPT-3.5 solutions across specified machine learning metrics".

Similarly, for the time comparison as "This approach minimizes the time (it takes less than 1 minute to generate a solution)".... did you compare with GPT 3.5? Please include time comparison metric as well.

If possible, please modify the Appendix C, and listings C1-C4. for at least 2 competitions, a side-by-side tabular format comparison, for --> "examples of code inferred by GPT-3.5 with two variations of task-describing prompts: one with and one without the automatically chosen best instruction." ….It will improve readability of the manuscript and experimental findings as well as help extended research community to better understand the "differentiation". Please make a table as [Left column] without and [Right column] with...........feel free to use any other presentable format ..

Additional comments

Authors are advised to release a public GitHub code repository upon acceptance of their research contribution for extended research community.

The research manuscript is interesting and research findings may add significant contribution.
I recommend major revision of the manuscript in its present form and resubmission is encouraged.

Reviewer 2 ·

Basic reporting

1. The paper is clear, informative, and structured. If the paper could discuss the direction of future work, it would be better.

2. The choice of colors in different figures can be consistent.

Experimental design

1. The experimental design has a clear description of the methods used.

Validity of the findings

1. The paper could also use other evaluation metrics to provide a more comprehensive assessment of the generated models.

2. The paper doesn't handle different types of ML tasks. There are lack of diverse task types being actually tested here.

3. One focus for me is interpretability. Further improving the interpretability and explainability of the framework's decision-making process could be beneficial.

Reviewer 3 ·

Basic reporting

I am impressed with the innovative approach your framework takes to bridge the gap between natural language descriptions and executable code in machine learning tasks. The use of a large language model to interpret and generate code solutions is both novel and practical. Your detailed methodology and extensive experimentation on the Kaggle platform provide a strong foundation for your findings. The paper is well-structured, with clear explanations of the problem, solution, and results. However, there are a few areas where improvements could be made. The introduction could benefit from a more detailed discussion of the knowledge gap you are addressing. The supplemental files also need more descriptive metadata for future readers.

Experimental design

no comment

Validity of the findings

no comment

Additional comments

no comment

---

## Round 0.2 · accepted · Accept

· Academic Editor

Accept

Thanks to the authors for their efforts to improve the work. This version successfully satisfied the reviewer. It may be accepted currently. Congrats!

Reviewer 2 ·

Basic reporting

My previous feedback has been addressed, and everything looks good now.

Experimental design

My previous feedback has been addressed, and everything looks good now.

Validity of the findings

My previous feedback has been addressed, and everything looks good now.